# Molecular Markers of Telomerase Complex for Patients with Pituitary Adenoma

**DOI:** 10.3390/brainsci12080980

**Published:** 2022-07-25

**Authors:** Greta Gedvilaite, Alvita Vilkeviciute, Brigita Glebauskiene, Loresa Kriauciuniene, Rasa Liutkeviciene

**Affiliations:** Neuroscience Institute, Medical Academy, Lithuanian University of Health Sciences, Eiveniu 2, LT-50161 Kaunas, Lithuania; alvita.vilkeviciute@lsmuni.lt (A.V.); brigita.glebauskiene@lsmuni.lt (B.G.); loresa.kriauciuniene@lsmuni.lt (L.K.); rasa.liutkeviciene@lsmuni.lt (R.L.)

**Keywords:** pituitary adenoma, telomerase complex, rs35073794

## Abstract

Pituitary adenoma (PA) is the most common benign tumor of the pituitary gland. The pathogenesis of most PA is considered as a multifactorial process, that involves genetic mutations, alterations in gene transcription, and epigenetic factors. Their interaction promotes tumorigenesis. The processes are increasingly focused on changes in telomere length. Our study enrolled 126 patients with PA and 368 healthy subjects. DNA samples from peripheral blood leukocytes were purified by the DNA salting-out method. The RT-PCR carried out SNPs and relative leukocyte telomere lengths (RLTL). ELISA determined the level of TEP1 in blood serum. Binary logistic regression revealed that *TERC* rs35073794 is likely associated with increased odds of PA development and macro-PA development. It is also associated with decreased odds of active PA, non-invasive PA, and PA without relapse development. Also, we discovered that PA patients with at least one G allele of the *TEP1* gene polymorphism rs1713418 have lower serum TEP1 levels than healthy individuals (*p* = 0.035). To conclude, the study revealed that *TERC* rs35073794 might be a potential biomarker for PA development.

## 1. Introduction

Pituitary adenomas (PA) are the most common benign tumors of the pituitary gland [1]. PAs are classified according to the primary origin of the cells and the type of hormones secreted. It is classified as nonfunctional if an adenoma does not secrete enough hormones to be detectable in the blood or cause clinical manifestations [1,2,3]. Prolactinomas account for 40–57% of all adenomas, followed by nonfunctional adenomas (28–37%), growth hormone-releasing adenomas (11–13%), and adrenocorticotropic hormone (ACTH)-related adenomas. PA that secretes follicle-stimulating hormone (FSH), luteinizing hormone (LH), or thyroid-stimulating hormone (TSH) is rare. Tumors are also categorized by size: if the tumor is 10 mm or larger, it is considered a macroadenoma; if it is smaller than 10 mm, it is considered a microadenoma. Microadenomas are slightly more common than macroadenomas (57.4% vs. 42.6%) [1,2]. Macro pituitary adenomas account for approximately 6–10% of all pituitary tumors. These adenomas are usually clinically dysfunctional and are more common in men. Symptoms are usually secondary to compression of nearby structures but may also be due to partial or complete hypopituitarism [4]. Pituitary tumors can cause various health problems depending on functional status, size, and invasion [5]. New molecular biomarkers could help diagnose the disease early and thus increase the survival rate of patients [6]. Molecular biomarkers act as an indicators of biological, pathological processes, or pharmacological responses that provide helpful information for diagnosing the disease and predicting its outcome [7].

Telomeres are nucleoprotein complexes at the ends of eukaryotic chromosomes. It is well known that telomeres shorten with age. The progressive shortening of telomeres leads to somatic cell aging, apoptosis, or oncogenic transformation, affecting human health and life expectancy. Shorter telomeres are associated with increased disease development and poor survival [8]. Most tumors secrete the enzyme telomerase, which is necessary for maintaining telomere length and thus for unlimited cell proliferation. Advances in understanding telomerase activity and telomere structure and the identification of telomerase and telomere-related proteins are opening new opportunities for therapeutic intervention [9]. Telomerase consists of telomerase RNA elements (TERC), telomerase reverse transcriptase (TERT), and telomerase-associated proteins [10]. TERC serves as a telomerase template to catalyze the attachment of telomeric DNA repeats to the 3’ ends of chromosomes. Telomerase dysfunction caused by mutations in the human *TERC* gene has been associated with many diseases, including tumors, pulmonary fibrosis, and early aging syndromes. However, the mechanisms by which these mutations cause telomerase dysfunction are poorly known [11]. It should also be noted that the telomerase RNA component (TERC) gene is involved in the early stages of tumor formation. TERC is a template for telomeric DNA synthesis. In addition, the gene is involved in processes such as catalysis, accumulation, and holoenzyme assembly [12]. Reactivation of *TERT* expression is a hallmark of cancer development and is observed in more than 80–90% of tumors. Therefore, a more precise characterization of telomerase regulation is attractive for developing new therapeutic targets and biomarkers in various pathologies [13]. When the *TERT* gene is overexpressed in normal cells, it can lead to a longer cell lifespan and transformation. Although telomerase activity is undetectable in most normal tissues, it is detected in approximately 90% of human tumors [14]. It is known that TEP1 may share common cellular functions that have physiological and/or pathological consequences. These associations in molecular mechanisms require further investigation [15,16]. Genetic variants of *TEP1* have already been studied concerning cancer risk, including the risk and prognosis of bladder and breast cancer [17]. However, there is no information on how it is related to PA.

Thus, this study aimed to determine the relationship between the molecular markers of telomerase complex (*TERC* rs12696304, rs35073794, *TEP1* rs1713418, rs1760904, and *TERT* rs2736098, rs401681) and the relative leukocyte telomeres length with the development of pituitary adenoma. Also, we aimed to investigate the relationship between serum TEP1 levels and pituitary adenoma development.

## 2. Materials and Methods

The study was carried out in the Department of Ophthalmology, Lithuanian University of Health Sciences. Kaunas Regional Biomedical Research Ethics Committee approved the study (No. BE-2-47, issued on 25 December 2016). All participants in the study signed the informed consent form. The subjects were divided into two groups:group I: patients with pituitary adenoma (*n* = 125) aged 19 to 80 years. The group consisted of 75 (59.5%) females and 51 (40.5%) males.

The PA group included PAs diagnosed and confirmed by magnetic resonance imaging (MRI); patients with good general health, aged 18 years and above, and with the absence of other tumors.

PA group were divided into subgroups by invasiveness, hormonal activity and relapse. Based on the histopathological view group was divided into invasive and non-invasive; based on hormone levels in blood serum, two groups were created: active and inactive PA. Relapse was diagnosed if the enlargement of a residual tumor or a new growth was noticed and documented on follow-up MRI. PA’s hormonal activity, recurrence [18] and invasiveness [19] were previously briefly described.

group II: healthy subjects (*n* = 368) aged 19 to 94 years. The group consisted of 217 (59.0%) females and 151 (41.0%) males.

The healthy controls group consisted of age and gender matched subjects having a good general health.

### 2.1. DNA Extraction and Genotyping

Venous blood samples were used for the DNA extraction and described earlier (19). Genotyping of single nucleotide polymorphisms (SNPs) of *TEP1* rs1713418, rs1760904, *TERC* rs12696304, rs35073794, and *TERT* rs2736098, rs401681 was carried out using the real-time polymerase chain reaction (RT-PCR) method. SNPs were determined using TaqMan^®^ Genotyping assays (Applied Biosystems, New York, NY, USA; Thermo Fisher Scientific, Inc., Waltham, MA, USA), C___8921332_10, C___1772362_20, C___407063_10, C___58097851_10, C___26414916_20, and C___1150767_20 according to manufacturer’s protocols by a StepOne Plus (Applied Biosystems).

### 2.2. Relative Leukocyte Telomeres Length Measurement

Relative leukocyte telomeres length (RLTL) was measured using the quantitative real-time PCR method Cawthon (2002) described. The amounts of telomere DNA fragments and the reference gene albumin were determined in 2 replicates. We performed RT-PCR to determine the relative length of leukocyte telomeres using a real-time PCR multiplier, StepOne Plus (Applied Biosystems, New York, NY, USA). The reference DNA was a mixture of two randomly selected test DNA samples. Positive control: DNA isolated from a commercial human cell line 1301 with an extra-long telomere (Sigma Aldrich, New York, NY, USA).

Primers:

Telg 5′-ACA CTA AGG TTT GGG TTT GGG TTT GGG TTT GGG TTA GTG T-3′

Telc 5′-TGT TAG GTA TCC CTA TCC CTA TCC CTA TCC CTA TCC CTA ACA-3′

Albd 5′-GCC CGG CCC GCC GCG CCC GTC CCG CCG GAA AAG CAT GGT CGC CTG TT-3′

Albu 5′-CGG CGG CGG GCG GCG CGG GCT GGG CGG AAA TGCTGC ACA GAA TCC TTG-3′

### 2.3. TEP1 Protein Measurement

TEP1 protein expression in blood serum was determined using ELISA. ELISA is an enzyme immunoassay for detecting and quantifying peptides, secreted proteins, hormones, and cytokines. ThermoFisher’s Human TEP1 ELISA Kit, based on solid-phase sandwich technology, was used to determine TEP1 protein levels in the study groups.

### 2.4. Statistical Analysis

Statistical analysis was performed using the SPSS/W 23.0 software (Statistical Package for the Social Sciences for Windows, Inc., Chicago, IL, USA). The frequencies of genotypes and alleles are presented in percentage. The hypothesis about the normal distribution of the measured trait values was tested using Kolmogorov-Smirnov and Shapiro-Wilk tests. Subjects’ characteristics did not meet the criteria of normality distribution; the following descriptive statistics characteristics were used: Median, interquartile range (IQR). The Fisher-Exact test was used to compare the distribution homogeneity for *TEP1* rs1713418, rs1760904, *TERC* rs12696304, rs35073794, *TERT* rs2736098, rs401681. After binary logistic regression analysis, PA development was estimated considering inheritance models and genotype combinations, giving an OR with a 95% confidence interval (CI). The Akaike Information Criterion (AIC) selected the best inheritance model, with the lowest value indicating the most appropriate model. A nonparametric Mann-Whitney U test compared different groups when the data distribution was not normal. A statistically significant difference was found when the *p*-value was less than 0.05.

## 3. Results

The study included 494 subjects divided into two groups: a control group (*n* = 368) and patients with pituitary adenoma (*n* = 126). Genotyping of *TEP1* rs1760904, rs1713418, *TERC* rs12696304, rs35073794, TERT rs2736098, and rs401681 polymorphisms was performed after the formation of the study groups. The patients with pituitary adenoma consisted of 126 subjects: 51 males (40.5%) and 75 females (59.5%). The median age of the patients was 53, and IQR was 23. The control group consisted of 368 subjects: 151 males (41%) and 217 females (59%). The median age of the control group was 57, and IQR 37. Relative leukocyte telomere length (RLTL) was determined in 52 subjects with pituitary adenoma and 226 healthy subjects. No statistically significant differences were found within the control and pituitary adenoma groups between sex, age, and relative leukocyte telomere length (*p* = 0.913, *p* = 0.615, and *p* = 0.901, respectively). The demographic data of the subjects are shown in Table 1.

### 3.1. TEP1 rs1760904, rs1713418, TERC rs12696304, rs35073794, TERT rs2736098, and rs401681 Associations with Pituitary Adenoma

The frequencies of genotypes and alleles of *TEP1* rs1760904, rs1713418, *TERC* rs12696304, rs35073794, *TERT* rs2736098, and rs401681 were performed within groups.

There were no statistically significant differences between the distribution of genotypes and alleles for patients with PA and the control group for the following single-nucleotide polymorphisms: *TEP1* rs1760904, rs1713418, *TERC* rs12696304, *TERT* rs2736098, rs401681 (Appendix A). Only *TERC* rs35073794 revealed statistically significant results between the groups. AG genotype was more frequent in patients with PA than in the control group (84.9 vs. 57.6, *p* < 0.001), while GG genotype was less frequent in PA than in the control group (0.8 vs. 42.4, *p* < 0.001) (Table 2).

Binary logistic regression revealed that *TERC* rs35073794 AG genotype under the codominant model, AG + AA genotype under the dominant model, AG genotype under the overdominant model, and G allele under additive model increased the odds of PA development by 79 times (OR:78.736; 95% CI: 10.872–570.221; *p* < 0.001), 92 times (OR: 91.981; 95% CI: 12.717–665.277; *p* < 0.001), 4 times (OR: 4.144; 95% CI: 2.439–7.040; *p* < 0.001), 114 times (OR: 114.329; 95% CI: 15.941–819.982; *p* < 0.001), respectively (Table 3).

Analysis of *TEP1* rs1760904, rs1713418, *TERC* rs12696304, *TERT* rs2736098, and rs401681 gene polymorphisms revealed no statistically significant differences between PA and control group.

### 3.2. TEP1 rs1760904, rs1713418, TERC rs12696304, rs35073794, TERT rs2736098, and rs401681 Associations with Pituitary Adenomas Activity

*TEP1*, *TERC*, and *TERT* genes single nucleotide polymorphisms were analyzed to evaluate the associations with pituitary adenoma activity, invasiveness, and relapse.

*TEP1* rs1713418 polymorphism analysis revealed statistically significant results when comparing active PA and control groups. It was found that the GG genotype in the group of active pituitary adenoma was statistically significantly lower than in the control group (6.0% vs. 16.3%, *p* = 0.045). Analysis revealed that *TERC* rs35073794 affects both–active and inactive pituitary adenoma. It was found that GG genotype is less frequent in the active PA than in the control group, as well as inactive PA groups than in the control group (1.5 vs. 42.4, *p* < 0.001; 9.1 vs. 42.4, *p* < 0.001, respectively). While AG genotype was found to be more frequent in the active PA than in the control group (79.1 vs. 57.6, *p* = 0.001) and in inactive PA than in the control group (90.9 vs. 57.6, *p* < 0.001) (Table 4). Analysis of the distribution of genotypes and alleles of the *TEP1* rs1760904, *TERC* rs12696304, *TERT* rs2736098, and rs401681 single nucleotide polymorphisms revealed no statistically significant differences between patients with inactive or active PA or the control group (Appendix A).

Binary logistic regression analysis showed that *TEP1* rs1713418 GG comparing with AA + AG, under recessive model, are associated with 3-fold increased odds of active PA development (OR 3.068; 95% CI: 1.076–8.748; *p* = 0.036). *TERC* rs35073794 AG genotype under codominant model, AG + AA genotype under dominant model, AG genotype under overdominant model and G allele under additive model decreased the odds of active PA development by 38 times (OR 0.026; 95% CI: 0.004–0.187; *p* < 0.001), 47 times (OR 0.021; 95% CI: 0.003–0.150; *p* < 0.001), 3 times (OR 0.359; 95% CI: 0.192–0.670; *p* = 0.001) and 91 time (OR 0.011; 95% CI: 0.002–0.082; *p* < 0.001), respectively. Also, *TERC* rs35073794 AG compared with AA + GG, under the overdominant model, is associated with 7-fold decreased odds of inactive PA development (OR 0.136; 95% CI: 0.053–0.349; *p* < 0.001) (Table 5). Analysis of *TEP1* rs1760904, *TERC* rs12696304, TERT rs2736098, rs401681 did not show statistically significant results.

### 3.3. TEP1 rs1760904, rs1713418, TERC rs12696304, rs35073794, TERT rs2736098, and rs401681 Associations with Pituitary Adenomas Invasiveness

When the distribution of genotypes and alleles of SNPs was analyzed between the invasive PA and control groups, it was found that *TEP1* rs1713418, the GG genotype was statistically significantly less frequent in the invasive PA group (7.4% vs. 16.3%, *p* = 0.041) (Table 6).

The analysis shows that *TERC* rs35073794 affects both–invasive and non-invasive PA. It was found that AG genotype is more frequent in the invasive PA than in the control group and non-invasive PA group than in the control group (86.4 vs. 57.6, *p* < 0.001; 81.8 vs. 57.6 *p* = 0.002, respectively) (Table 6). Also, *TERT* rs2736098 TT genotype was more frequent in invasive PA than in the control group (16.0% vs. 7.6%, *p* = 0.017) (Table 6).

Analysis of *TEP1* rs1760904, *TERC* rs12696304, and *TERT* rs401681 revealed no statistically significant differences between patients with invasive or non-invasive PA and the control group (Appendix A).

Binary logistic regression between invasive or non-invasive PA and control groups was performed. It revealed that *TEP1* rs1713418 GG compared with AA + AG, under the recessive model, is associated with 2-fold increased odds of invasive PA development (OR 2.435; 95% CI: 11.014–5.849; *p* = 0.047). *TERC* rs35073794 AG compared with AA + GG, under the overdominant model, is associated with 5-fold decreased odds of invasive PA development (OR 0.214; 95% CI: 0.109–0.417; *p* < 0.001). *TERT* rs2736098 TT, compared with CC, is associated with 2-fold decreased odds of invasive PA development (OR 0.439; 95% CI: 0.211–0.912; *p* = 0.027). Also, under the recessive model, TT compared with CC + CT is associated with 2-fold decreased odds of invasive PA development (OR 0.431; 95% CI: 0.212–0.874; *p* = 0.020) (Table 7). Also, *TERC* rs35073794 AG genotype under the codominant model, AG + AA genotype under the dominant model, AG genotype under the overdominant model, and G allele under additive model decreased the odds of non-invasive PA development by 26 times (OR: 0.038; 95% CI: 0.005–0.278; *p* = 0.001), 31 time (OR: 0.032; 95% CI: 0.004–0.232; *p* = 0.001), 3 times (OR: 0.302; 95% CI: 0.137–0.668; *p* = 0.003), 63 times (OR: 0.016; 95% CI: 0.002–0.116; *p* < 0.001), respectively (Table 7). *TEP1* rs1760904, *TERC* rs12696304, and *TERT* rs401681 did not show statistically significant results.

### 3.4. TEP1 rs1760904, rs1713418, TERC rs12696304, rs35073794, TERT rs2736098, and rs401681 Associations with Pituitary Adenomas Relapse

*TEP1* rs1713418 GG genotype was statistically less frequent in the PA without relapse group than in the control group (6.5% vs. 16.3%, *p* = 0.017). *TERC* rs35073794 affects both–PA with and without relapse. It was found that AG genotype is more frequent in the PA with relapse than in the control group and PA without relapse group than in the control group (90.0 vs. 57.6, *p* < 0.001; 82.6 vs. 57.6 *p* < 0.001, respectively). Also, the *TERT* rs2736098 TT genotype was more frequent in PA without relapse group than in the control group (15.2% vs. 7.6%, *p* = 0.026) (Table 8). Analysis of the genotype and allele distributions of *TEP1* rs1760904, *TERC* rs12696304, and *TERT* rs401681 did not reveal statistically significant differences between the groups (Appendix A).

Binary logistic regression analysis was performed within PA with or without relapse and control group subjects. It revealed that *TERC* rs35073794 AG compared with AA + GG, under the overdominant model, is associated with 7-fold decreased odds of PA with relapse development (OR 0.151; 95% CI: 0.045–0.507; *p* = 0.002). *TEP1* rs1713418 GG compared with AA + AG, under the recessive model, is associated with 3-fold increased odds of PA without relapse development (OR 2.792; 95% CI: 1.167–6.682; *p* = 0.021). *TERC* rs35073794 AG genotype under the codominant model, AG + AA genotype under the dominant model, AG genotype under the overdominant model, and G allele under additive model decreased the odds of PA without relapse development by 56 times (OR: 0.018; 95% CI: 0.002–0.130; *p* < 0.001), 67 times (OR: 0.015; 95% CI: 0.002–0.108; *p* < 0.001), 3 times (OR: 0.286; 95% CI: 0.161–0.510; *p* < 0.001), 100 times (OR: 0.010; 95% CI: 0.001–0.074; *p* < 0.001), respectively. *TERT* rs2736098 TT, compared with CC, is associated with 2-fold decreased odds of PA without relapse development (OR 0.471; 95% CI: 0.232–0.956; *p* = 0.037). Also, under the recessive model, TT compared with CC + CT is associated with 2-fold decreased odds of invasive PA development (OR 0.459; 95% CI: 0.231–0.912; *p* = 0.026) (Table 9).

### 3.5. TEP1 rs1760904, rs1713418, TERC rs12696304, rs35073794, TERT rs2736098, and rs401681 Associations with Micro and Macro Pituitary Adenomas

The distribution of genotypes and alleles of SNPs was analyzed between the micro/macro PA and control groups. It was found that *TERC* rs35073794 affects both–macro and micro PA. Analysis showed that AG genotype is more frequent in the macro PA than in the control group and micro PA group than in the control group (84.8 vs. 57.6, *p* < 0.001; 82.9 vs. 57.6 *p* = 0.002, respectively) (Table 10). Also, *TERT* rs2736098 TT genotype was more frequent in macro PA than in the control group (16.5% vs. 7.6%, *p* = 0.013), as well as each T allele was more frequent in micro PA than in the control group (31.6% vs. 23.8%, *p* = 0.039) (Table 10).

*TEP1* rs1760904, rs1713418, *TERC* rs12696304, and *TERT* rs401681 revealed no statistically significant differences between invasive or non-invasive PA patients and the control group (Appendix A).

Binary logistic regression analysis was performed within micro/macro PA and control group subjects. It revealed that *TERC* rs35073794 AG compared with AA + GG, under the overdominant model, is associated with 4-fold decreased odds of micro PA (OR 0.280; 95% CI: 0.121–0.648; *p* = 0.003). *TERC* rs35073794 has an opposite effect on macro PA: AG genotype under the codominant model, AG + AA genotype under the dominant model, AG genotype under the overdominant model, and G allele under additive model increased the odds of macro PA development by 49 times (OR: 49.302; 95% CI: 6.771–358.990; *p* < 0.001), 57 times (OR: 57.393; 95% CI: 7.899–417.078; *p* < 0.001), 4 times (OR: 4.108; 95% CI: 2.149–7.856; *p* < 0.001), 82 times (OR: 83.234; 95% CI: 11.522–601.271; *p* < 0.001), respectively. Also, *TERT* rs2736098 TT genotype under the codominant model increases the odds of macro PA development by 2 times, as well as under the recessive model (OR: 2.443; 95% CI: 1.170–5.099; *p* = 0.017; OR: 2.392; 95% CI: 1.177–4.858; *p* = 0.016, respectively) (Table 11).

### 3.6. Serum TEP1 Levels

Serum TEP1 levels were measured in patients with pituitary adenoma (*n* = 20) and control (*n* = 31) groups, but no statistically significant difference was found (median (IQR): 236 (127) vs. 269 (200), *p* = 0.370). Results are shown in Appendix A.

A comparison of serum TEP1 levels between different genotypes for selected single nucleotide polymorphisms was performed. PA patients with at least one G allele of the *TEP1* gene polymorphism rs1713418 had lower serum TEP1 levels than healthy individuals (*p* = 0.035) (Table 12). No statistically significant differences were found between *TEP1* rs1760904, *TERC* rs12696304, rs35073794, *TERT* rs2736098, and rs401681 (Appendix A).

### 3.7. Relative Leukocyte Telomeres Length Associations with Pituitary Adenoma

Relative leukocyte telomeres length (RLTL) was measured for 52 patients with pituitary adenoma and 226 control group subjects. We found that there was no statistically significant difference in RLTL between the PA group and the control group (median (IQR): 0.645 (1.112) vs. 0.603 (0.635), *p* = 0.901). The results are shown in Appendix A.

A comparison of RLTL between different genotypes was also performed for all studied polymorphisms. Unfortunately, there were no statistically significant results comparing PA and control groups (Appendix A).

## 4. Discussion

To the best of our knowledge, no studies have been performed to analyze the association of *TEP1* rs1760904, rs1713418, *TERC* rs12696304, rs35073794, *TERT* rs2736098, rs401681 gene polymorphisms with PA development and invasiveness, activity, or relapse. It should be noted that genetic variants that affect telomere length, telomerase activation, and telomeric protein configuration may cause functional changes responsible for cancer development and further tumor growth. Cancer Genome Association Studies have shown that single-nucleotide polymorphisms (SNPs) in telomere-associated genes are associated with a risk of developing various tumors [20]. In the literature on the selected polymorphisms, only a possible association with tumors of other localizations has been described [21,22,23,24,25,26,27].

Litviakov and co-authors found that *TEP1* rs1760904 is associated with different types of chromosomal abnormalities [28]. *TEP1* SNP (rs1760897) has recently been associated with an increased risk of bladder cancer [21]. Gu and co-authors found that the distribution of *TEP1* rs1760904 polymorphism genotypes between the control and prostate cancer groups was statistically significant (*p* = 0.012). The researchers also found that *TEP1* rs1760904 AG and AG/AA genotypes were statistically significantly associated with a reduced risk of prostate cancer compared to the GG genotype (*p* = 0.003; *p* = 0.005, respectively). Moreover, a significant interaction was found between the *TEP1* rs1713418 polymorphism and age, with AG/GG genotypes associated with a 32% increased risk of prostate cancer in younger subjects and 29% reduced risk in older subjects [20]. Our study confirms that *TEP1* rs1713418 affects tumorogenesis. GG genotype in the group of active pituitary adenoma was statistically significantly less frequent than in the control group (6.0% vs. 16.3%, *p* = 0.045). Under the recessive model, GG genotype compared with AA + AG is associated with 3-fold increased odds of active PA development (OR 3.068; 95% CI: 1.076–8.748; *p* = 0.036). When the distribution of genotypes and alleles of SNPs was analyzed between the invasive PA and control groups, it also revealed that the GG genotype was statistically significantly less frequent in the invasive PA group (7.4% vs. 16.3%, *p* = 0.041). AG genotype is more frequent in the invasive PA than in the control group and non-invasive PA group than in the control group (86.4 vs. 57.6, *p* < 0.001; 81.8 vs. 57.6 *p* = 0.002, respectively). *TEP1* rs1713418 GG compared with AA + AG, under the recessive model, is associated with 2-fold increased odds of invasive PA development (OR 2.435; 95% CI: 11.014–5.849; *p* = 0.047). *TEP1* rs1713418 GG genotype was statistically less frequent in the PA without relapse group than in the control group (6.5% vs. 16.3%, *p* = 0.017). *TEP1* rs1713418 GG compared with AA + AG, under the recessive model, is associated with 3-fold increased odds of PA without relapse development (OR 2.792; 95% CI: 1.167–6.682; *p* = 0.021).

The *TERT* gene is thought to be involved in an apoptotic response, and its level of expression affects the susceptibility of cancer cell lines to anticancer drugs. Whole-genome association studies have identified the *TERT* gene rs401681 as one of the most significant cancer-associated SNPs [29]. A study made by Ma and co-authors discusses the relationship between *TERTp* mutation and telomere length. Gene mutation was positively and significantly correlated with telomere shortening (*p* = 0,032) [30]. However, our study did not reveal any differences between SNPs and RLTL in patients with PA with the control group subjects.

It should be noted that longer telomeres may slow aging, increase cell proliferative potential, and accelerate cancer progression [31]. Veryan et al. found that each rarer *TERC* rs12696304 allele was associated with a shorter mean telomere length in patients [32]. Recent genetic variants identified by Beatrice and co-authors in the *hTERC* region showing an association between rs12696304 and RLTL were confirmed at a median age of 60 (OR 0.25 (0.12–0.54)) [33].

Regarding tumor formation, Huang et al. confirm that rs35073794 statistically significantly reduced the development of hepatocellular carcinoma according to the additive model OR 0.82 (0.68–1.00), *p* = 0.049 [22]. Wu and co-authors found that *TERC* rs35073794 is associated with an increased risk of renal cell carcinoma (RCC) in a codominant model (OR 2.61 (1.01–6.76), *p* = 0.045). Also, polymorphism is positively correlated with age over 55 years (OR 3.27 (1.08–9.93), *p* = 0.031) [23]. Researchers have found that rs35073794 can be used as a diagnostic and prognostic marker in clinical trials in patients with renal cell carcinoma [23]. The findings in our study reveal that *TERC* rs35073794 is strongly associated with the development of pituitary adenoma. *TERC* rs35073794 AG genotype increased the odds of PA development. Therefore, decreased the odds of active PA development and the odds of non-invasive PA development also decreased the odds of PA without relapse development. *TERC* rs35073794 has an opposite effect on macro-PA: increased the odds of macro-PA development.

Researchers found that the *TERT* rs2736098 genotype variants (GA/AA) were statistically significantly associated with an increased risk of cancer (GA/AA and GG: OR 1.14 (1.04–1.25)), and in subsequent analyzes, rs2736098 is associated with an increased risk of lung cancer (OR 1.18 (1.07–1.29)) and risk of hepatocellular carcinoma (OR 1.38 (1.20–1/59)) [24]. Logistic regression showed that subjects with the A allele or the AA genotype had a statistically significant increased risk of lung cancer compared with those with the G allele or the GG genotype (A and G: OR 1.21 (1.02–1.43), *p* = 0.028; AA and GG: OR 1.48 (1.05–2.09), *p* = 0.025), and this association was stronger between adenocarcinoma cases (AA and GG: OR 1, 67 (1.12–2.50), *p* = 0.013; A and G: OR 1.28 (1.05–1.57), *p* = 0.016) [25]. Hashemi and co-authors found that *TERT* rs2736098 polymorphism genotype increased the risk of breast cancer (OR 1.80 (1.12–2.88), *p* = 0.017, HP versus AA; OR 1.80 (1.06–3.06), *p* = 0,033, GG vs. AA; GS = 1.87 (1.19–2.94), *p* = 0.006, AG + GG versus AA) [26]. A significant association between rs401681 polymorphism and cancer risk has also been observed [27].

In our study, analyzing the *TERT* rs2736098 TT genotype distribution was more frequent in invasive PA than in the control group (*p* = 0.017). Also, the *TERT* rs2736098 TT genotype was more frequent in PA without relapse group than in the control group (*p* = 0.026). *TERT* rs2736098 TT genotype was more frequent in macro-PA than in the control group (*p* = 0.013), and each T allele was more frequent in micro-PA than in the control group (*p* = 0.039).

Telomere-associated genes play an essential role in carcinogenesis and prostate cancer progression. However, it has not been fully elucidated whether genetic alterations in telomere-related genes are associated with HA progression [20]. We compared serum TEP1 levels between groups and genotypes for all polymorphisms. PA patients with at least one G allele of the *TEP1* gene rs1713418 polymorphism (*p* = 0.035) had lower serum TEP1 concentrations than healthy subjects. According to the researchers, higher TEP1 expression correlates with telomerase activity in cancer cells [34]. Also, evidence that selective depletion of TEP1 restores the biological effects observed in miR-380-5p-transfected cells suggests the possibility that the protein may be a target of miRNA [35].

Boresowicz et al. focused on the relationship between telomere length and the abnormalities of *TERT* gene, TERT expression, and clinicopathological features in PA patients. However, results revealed that telomerase abnormalities do not play a significant role in the pathogenesis of pituitary tumors [36]. It explains our study findings within telomere length and telomerase complex.

Thus, the research and results link the markers regulating the tumor process and the telomerase complex *TEP1* rs1760904, rs1713418, *TERC* rs12696304, rs35073794, *TERT* rs2736098, rs401681, which help explain the factors leading to the development of telocytes. It is appropriate to develop these studies in further research. Interpretation of this association must await the results of similar studies conducted on other populations, and additional research is needed to evaluate the clinical relevance of this interaction.

## 5. Conclusions

*TERC* rs35073794 may be considered as a potential biomarker for PA.

## Figures and Tables

**Table 1 brainsci-12-00980-t001:** Demographic characteristics.

Characteristics		Group		*p*-Value
		Pituitary Adenoma Group N = 126	Control Group N = 368	
Gender	Males, N (%)	51 (40.5)	151 (41.0)	0.913
	Females, N (%)	75 (59.5)	217 (59.0)	
Age, Median (IQR)		53 (23)	57 (37)	0.615 *
Invasiveness			-	-
Yes		81(64.3)		
No		44 (34.9)		
Not determned		1 (1)		
Relapse			-	-
Yes		30 (23.8)		
No		92 (73)		
Not determined		4 (3.2)		
Activity			-	-
Yes		67 (53.2)		
No		55 (43.7)		
Not determined		4 (3.2)		

* Mann Whitney U test.

**Table 2 brainsci-12-00980-t002:** Genotype and allele frequencies of single nucleotide polymorphism (*TERC* rs35073794) within PA and control groups.

Gene, SNP	Genotype, Allele	PA Group, N (%)	Control Group, N (%)	*p*-Value
*TERC* rs35073794	GG	1 (0.8) ^1^	156 (42.4) ^1^	<0.001 <0.001
	AG	107 (84.9) ^2^	212 (57.6) ^2^	
	AA	18 (14.3)	0 (0.0)	
	Total	126 (100)	368 (100)	
	Allele			
	G	109 (43.3)	524 (71.2)	
	A	143 (56.7)	212 (28.8)	

^1^ *p* < 0.001 (GG vs. AA + AG); ^2^ *p* < 0.001 (AG vs. GG + AA).

**Table 3 brainsci-12-00980-t003:** Binary logistic regresion analysis within patients with pituitary adenoma and control group subjects.

Model	Genotype/Allele	OR (95% CI)	*p*-Value	AIC
*TERC* rs35073794				
Codominant	AG vs. GG	78.736 (10.872–570.221)	<0.001	423.120
	AA vs. GG	2.5208 × 10³ (0.000- -)	0.998	
Dominant	AG + AA vs. GG	91.981 (12.717–665.277)	<0.001	458.571
Recessive	AA vs. GG + AG	5,504,580,946.293 (0.000- -)	0.998	511.788
Overdominant	AG vs. AA + GG	4.144 (2.439–7.040)	<0.001	529.245
Additive	G	114.329 (15.941–819.982)	<0.001	421.856

**Table 4 brainsci-12-00980-t004:** *TEP1* and *TERC* genes single nucleotide polymorphisms Frequencies of genotypes and alleles within active or inactive pituitary adenoma and control groups.

Gene, SNP	Genotype, Allele	Control Group, N (%)	Active PA Group, N (%)	*p*-Value	Inactive PA Group, N (%)	*p*-Value
*TEP1* rs1713418	AA	136 (37.0)	23 (34.3)	0.046 0.457	22 (40.0)	0.776 0.576
	AG	172 (46.7)	40 (59.7)		26 (47.3)	
	GG	60 (16.3) ^1^	4 (6.0) ^1^		7 (12.7)	
	Total	368 (100)	67 (100)		55 (100)	
	Allele					
	A	444 (60.3)	86 (64.2)		70 (63.6)	
	G	292 (39.7)	48 (35.8)		40 (36.4)	
*TERC* rs35073794	GG	156 (42.4) ^2,4^	1 (1.5) ^2^	<0.001 <0.001	5 (9.1) ^4^	<0.001 <0.001
	AG	212 (57.6) ^3,5^	53 (79.1) ^3^		50 (90.9) ^5^	
	AA	0 (0.0)	13 (19.4)		5 (9.1)	
	Total	368 (100)	67 (100)		55 (100)	
	Allele					
	G	524 (71.2)	55 (41.0)		60 (50.0)	
	A	212 (28.8)	79 (59.0)		60 (50.0)	

^1^*p* = 0,045 (GG vs. AA + GA); ^2^ *p* < 0.001 (GG vs. AA + AG); ^3^ *p* = 0.001 (AG vs. GG + AA); ^4^ *p* < 0.001 (GG vs. AA + AG); ^5^
*p* < 0.001 (AG vs. GG + AA).

**Table 5 brainsci-12-00980-t005:** Binary logistic regression analysis within active or inactive PA and control group subjects.

Model	Genotype/Allele	OR (95% CI)	*p*-Value	AIC
Active PA				
*TEP1* rs1713418				
Codominant	AG vs. AA	0.727 (0.415–1.273)	0.265	370.706
	GG vs. AA	2.537 (0.841–7.654)	0.099	
Dominant	AG + GG vs. AA	0.892 (0.516–1.541)	0.681	375.603
Recessive	GG vs. AA + AG	3.068 (1.076–8.748)	0.036	369.971
Overdominant	AG vs. AA + GG	0.592 (0.349–1.006)	0.053	371.947
Additive	A	1.183 (0.803–1.743)	0.395	375.043
*TERC* rs35073794				
Codominant	AG vs. GG	0.026 (0.004–0.187)	<0.001	281.319
	AA vs. GG	0.000 (0.000- -)	0.998	
Dominant	AG + AA vs. GG	0.021 (0.003–0.150)	<0.001	318.836
Recessive	AA vs. GG + AG	0.000 (0.000- -)	0.998	324.825
Overdominant	AG vs. AA + GG	0.359 (0.192–0.670)	0.001	363.936
Additive	G	0.011 (0.002–0.082)	<0.001	280.958
Inactive PA				
*TERC* rs35073794				
Codominant	AG vs. GG	0.000 (0.000- -)	0.995	259.418
	AA vs. GG	0.000 (0.000- -)	0.998	
Dominant	AG + AA vs. GG	0.000 (0.000- -)	0.995	273.592
Recessive	AA vs. GG + AG	0.000 (0.000- -)	0.999	308.111
Overdominant	AG vs. AA + GG	0.136 (0.053–0.349)	<0.001	301.980
Additive	G	0.000 (0.000- -)	0.995	257.418

**Table 6 brainsci-12-00980-t006:** *TEP1*, *TERC*, and *TERT* genes single nucleotide polymorphisms frequencies of genotypes and alleles within invasive or non-invasive pituitary adenoma and control groups.

Gene, SNP	Genotype, Allele	Control Group, N (%)	Invasive PA Group, N (%)	*p*-Value	Non-Invasive PA Group, N (%)	*p*-Value
*TEP1* rs1713418	AA	136 (37.0)	30 (37.0)	0.098 0.289	15 (34.1)	0.772 0.986
	AG	172 (46.7)	45 (55.6)		23 (52.3)	
	GG	60 (16.3) ^1^	6 (7.4) ^1^		6 (13.6)	
	Total	368 (100)	81 (100)		44 (100)	
	Allele					
	A	444 (60.3)	105 (64.8)		53 (60.2)	
	G	292 (39.7)	57 (35.2)		35 (39.8)	
*TERC* rs35073794	GG	156 (42.4)	0 (0.0)	<0.001 <0.001	1 (2.3)	<0.001 <0.001
	AG	212 (57.6) ^2,3^	70 (86.4) ^2^		36 (81.8) ^3^	
	AA	0 (0.0)	11 (13.6)		7 (15.9)	
	Total	368 (100)	81 (100)		44 (100)	
	Allele					
	G	524 (71.2)	70 (43.2)		38 (43.2)	
	A	212 (28.8)	92 (56.8)		50 (56.8)	
*TERT* rs2736098	CC	221 (60.1)	45 (55.6)	0.057 0.085	26 (59.1)	0.963 0.986
	CT	119 (32.3)	23 (28.4)		15 (34.1)	
	TT	28 (7.6) ^4^	13 (16.0) ^4^		3 (6.8)	
	Total	368 (100)	81 (100)		44 (100)	
	Allele					
	C	561 (76.2)	113 (69.8)		67 (76.1)	
	T	175 (23.8)	49 (30.2)		21 (23.9)	

^1^*p* = 0.041 (GG vs. AA + AG); ^2^ *p* < 0.001 (AG vs. AA + GG); ^3^ *p* = 0.002 (AG vs. AA + GG); ^4^ *p* = 0.017 (TT vs. CC + CT).

**Table 7 brainsci-12-00980-t007:** Binary logistic regression analysis within invasive or non-invasive PA and control group subjects.

Model	Genotype/Allele	OR (95% CI)	*p*-Value	AIC
Invasive PA				
*TEP1* rs1713418				
Codominant	AG vs. AA	0.843 (0.504–1.409)	0.515	422.616
	GG vs. AA	2.206 (0.872–5.578)	0.095	
Dominant	AG + GG vs. AA	1.003 (0.610–1.651)	0.989	425.857
Recessive	GG vs. AA + AG	2.435 (1.014–5.849)	0.047	421.042
Overdominant	AG vs. AA + GG	0.702 (0.423–1.139)	0.152	423.790
Additive	A	1.216 (0.850–1.740)	0.285	424.699
*TERC* rs35073794				
Codominant	AG vs. GG	0.000 (0.000- -)	0.995	320.054
	AA vs. GG	0.000 (0.000- -)	0.997	
Dominant	AG + AA vs. GG	0.000 (0.000- -)	0.995	347.488
Recessive	AA vs. GG + AG	0.000 (0.000- -)	0.998	386.885
Overdominant	AG vs. AA + GG	0.214 (0.109–0.417)	<0.001	401.155
Additive	G	0.000 (0.000- -)	0.995	318.054
*TERT* rs2736098				
Codominant	CT vs. CC	1.054 (0.608–1.825)	0.853	422.844
	TT vs. CC	0.439 (0.211–0.912)	0.027	
Dominant	CT + TT vs. CC	0.831 (0.512–1.351)	0.456	425.304
Recessive	TT vs. CC + CT	0.431 (0.212–0.874)	0.020	420.879
Overdominant	CT vs. CC + TT	1.205 (0.709–2.048)	0.490	425.372
Additive	C	0.752 (0.531–1.067)	0.111	423.376
Non-invasive PA				
*TERC* rs35073794				
Codominant	AG vs. GG	0.038 (0.005–0.278)	0.001	221.561
	AA vs. GG	0.000 (0.000- -)	0.999	
Dominant	AG + AA vs. GG	0.032 (0.004–0.232)	0.001	245.495
Recessive	AA vs. GG + AG	0.000 (0.000- -)	0.999	249.592
Overdominant	AG vs. AA + GG	0.302 (0.137–0.668)	0.003	271.385
Additive	G	0.016 (0.002–0.116)	<0.001	221.340

**Table 8 brainsci-12-00980-t008:** *TEP1*, *TERC*, and *TERT* genes single nucleotide polymorphisms frequencies of genotypes and alleles within pituitary adenoma with relapse or without relapse and control groups.

Gene, SNP	Genotype, Allele	Control Group, N (%)	PA Group with Relapse, N (%)	*p*-Value	PA Group without Relapse, N (%)	*p*-Value
*TEP1* rs1713418	AA	136 (37.0)	12 (40.0)	0.932 0.838	33 (35.9)	0.035 0.279
	AG	172 (46.7)	13 (43.3)		53 (57.6)	
	GG	60 (16.3) ^1^	5(16.7)		6 (6.5) ^1^	
	Total	368 (100)	30 (100)		92 (100)	
	Allele					
	A	444 (60.3)	37 (61.7)		119 (64.7)	
	G	292 (39.7)	23 (38.3)		65 (35.3)	
*TERC* rs35073794	GG	156 (42.4)	0 (0.0)	<0.001 <0.001	1 (1.1)	<0.001 <0.001
	AG	212 (57.6) ^2,3^	27 (90.0) ^2^		76 (82.6) ^3^	
	AA	0 (0.0)	3 (10.0)		15 (16.3)	
	Total	368 (100)	30 (100)		92 (100)	
	Allele					
	G	524 (71.2)	27 (45.0)		78 (42.4)	
	A	212 (28.8)	33 (55.0)		106 (57.6)	
*TERT* rs2736098	CC	221 (60.1)	16 (53.3)	0.6910.614	52 (56.5)	0.0740.118
	CT	119 (32.3)	12 (40.0)		26 (28.3)	
	TT	28 (7.6) ^4^	2 (6.7)		14 (15.2) ^4^	
	Total	368 (100)	30 (100)		92 (100)	
	Allele					
	C	561 (76.2)	44 (73.3)		130 (70.7)	
	T	175 (23.8)	16 (26.7)		54 (29.3)	

^1^*p* = 0.017 (GG vs. AA + AG); ^2^ *p* < 0.001 (AG vs. AA + GG); ^3^ *p* < 0.001 (AG vs. AA + GG); ^4^ *p* = 0.026 (TT vs. CC + CT).

**Table 9 brainsci-12-00980-t009:** Binary logistic regression analysis within PA with or without relapse and control group subjects.

Model	Genotype/Allele	OR (95% CI)	*p*-Value	AIC
PA with relapse				
*TERC* rs35073794				
Codominant	AG vs. GG	0.000 (0.000- -)	0.995	202.582
	AA vs. GG	0.000 (0.000- -)	0.999	
Dominant	AG + AA vs. GG	0.000 (0.000- -)	0.995	183.381
Recessive	AA vs. GG + AG	0.000 (0.000- -)	0.999	198.996
Overdominant	AG vs. AA + GG	0.151 (0.045–0.507)	0.002	202.347
Additive	G	0.000 (0.000- -)	0.995	170.582
PA without relapse				
*TEP1* rs1713418				
Codominant	AG vs. AA	0.787 (0.483–1.285)	0.339	456.762
	GG vs. AA	2.426 (0.966–6.097)	0.059	
Dominant	AG + GG vs. AA	0.954 (0.593–1.535)	0.847	462.333
Recessive	GG vs. AA + AG	2.792 (1.167–6.682)	0.021	455.687
Overdominant	AG vs. AA + GG	0.646 (0.407–1.024)	0.063	458.883
Additive	A	1.211 (0.860–1.705)	0.272	461.151
*TERC* rs35073794				
Codominant	AG vs. GG	0.018 (0.002–0.130)	<0.001	348.507
	AA vs. GG	0.000 (0.000- -)	0.998	
Dominant	AG + AA vs. GG	0.015 (0.002–0.108)	<0.001	384.459
Recessive	AA vs. GG + AG	0.000 (0.000- -)	0.998	411.991
Overdominant	AG vs. AA + GG	0.286 (0.161–0.510)	<0.001	440.861
Additive	G	0.010 (0.001–0.074)	<0.001	347.599
*TERT* rs2736098				
Codominant	CT vs. CC	1.077 (0.640–1.813)	0.780	459.722
	TT vs. CC	0.471 (0.232–0.956)	0.037	
Dominant	CT + TT vs. CC	0.865 (0.545–1.372)	0.537	461.991
Recessive	TT vs. CC + CT	0.459 (0.231–0.912)	0.026	457.800
Overdominant	CT vs. CC + TT	1.213 (0.733–2.007)	0.452	461.795
Additive	C	0.781 (0.559–1.091)	0.148	460.319

**Table 10 brainsci-12-00980-t010:** *TERC*, and *TERT* genes single nucleotide polymorphisms frequencies of genotypes and alleles within macro or micro pituitary adenoma and control groups.

Gene, SNP	Genotype, Allele	Control Group, N (%)	Macro PA, N (%)	*p*-Value	Micro PA, N (%)	*p*-Value
*TERC* rs35073794	GG	156 (42.4)	1 (1.3)	<0.001 <0.001	0 (0.0)	<0.001 <0.001
	AG	212 (57.6) ^1,2^	67 (84.8) ^1^		34 (82.9) ^2^	
	AA	0 (0.0)	11 (13.9)		7 (17.1)	
	Total	368 (100)	79 (100)		44 (100)	
	Allele					
	G	524 (71.2)	69 (43.7)		34 (41.5)	
	A	212 (28.8)	89 (56.3)		48 (58.5)	
*TERT* rs2736098	CC	221 (60.1)	42 (53.2)	0.046 0.039	24 (58.5)	0.973 0.902
	CT	119 (32.3)	24 (30.4)		14 (34.1)	
	TT	28 (7.6) ^3^	13 (16.5) ^3^		3 (7.3)	
	Total	368 (100)	79 (100)		41 (100)	
	Allele					
	C	561 (76.2)	108 (68.4)		62 (75.6)	
	T	175 (23.8)	50 (31.6)		20 (24.4)	

^1^*p* < 0.001 (AG vs. AA + GG); ^2^ *p* = 0.002 (AG vs. AA + GG); ^3^ *p* = 0.013 (TT vs. CC + CT).

**Table 11 brainsci-12-00980-t011:** Binary logistic regression analysis within micro or macro PA and control group subjects.

Model	Genotype/Allele	OR (95% CI)	*p*-Value	AIC
Micro PA				
*TERC* rs35073794				
Codominant	AG vs. GG	0.000 (0.000- -)	0.995	201.638
	AA vs. GG	0.000 (0.000- -)	0.998	
Dominant	AG + AA vs. GG	0.000 (0.000- -)	0.995	226.190
Recessive	AA vs. GG + AG	0.000 (0.000- -)	0.999	235.006
Overdominant	AG vs. AA + GG	0.280 (0.121–0.648)	0.003	257.403
Additive	G	0.000 (0.000- -)	0.995	199.638
Macro PA				
*TERC* rs35073794				
Codominant	AG vs. GG	49.302 (6.771–358.990)	<0.001	323.701
	AA vs. GG	-	-	
Dominant	AG + AA vs. GG	57.393 (7.899–417.078)	<0.001	351.798
Recessive	AA vs. GG + AG	-	-	-
Overdominant	AG vs. AA + GG	4.108 (2.149–7.856)	<0.001	396.054
Additive	G	83.234 (11.522–601.271)	<0.001	322.740
*TERT* rs2736098				
Codominant	CT vs. CC	1.061 (0.613–1.837)	0.832	415.617
	TT vs. CC	2.443 (1.170–5.099)	0.017	
Dominant	CT + TT vs. CC	1.324 (0.812–2.159)	0.260	417.702
Recessive	TT vs. CC + CT	2.392 (1.177–4.858)	0.016	413.662
Overdominant	CT vs. CC + TT	0.913 (0.539–1.546)	0.735	418.850
Additive	C	1.408 (0.992–2.000)	0.055	415.400

**Table 12 brainsci-12-00980-t012:** Frequencies of genotypes and serum TEP1 levels.

Genotype	Serum TEP1 Level		*p*-Value *
	PA Group Median (IQR)	Control Group Median (IQR)	
*TEP1* rs1713418			
AA	265.89 (108.21)	297.08 (204.90)	0.622
AG + GG	233.45 (132.81)	348.09 (324.42)	0.035

* Mann-Whitney U test.

## Data Availability

Data will be provided if a request is made by editors, reviewers, or scientists.

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
