# Peer review of "Molecular Markers of Telomerase Complex for Patients with Pituitary Adenoma"

_brainsci, 2022, doi:10.3390/brainsci12080980_

Round 1
Reviewer 1 Report
The article “TEP1, TERC, and TERT genetic variants in the development of Pituitary Adenoma’ describe the attempt of the evaluation of role of telomerase related SNPs in pituitary tumors. This issue was not explored previously. Appropriate methods were used for the study however the patients group is relatively small, especially in analysis of SNP associations in patients subgroups.
Comments
1. The main concern with this study is that it is focused on pituitary tumors as homogenous group. In fact particular subtypes of PitNEts are notably different in clinical manifestation, and molecular pathogenic mechanism including completely different mutations profile. Vast majority of the current and recently published studies on pituitary tumors take classification of PitNETs into account, which is definitely justified.
2. It is completely unknown what are diagnosis of the patients included in this study. Ay patients clinical /diagnostic characteristics are presented. Without this information it is unclear how to generalize the results of this study.
3. The following parameters were included in the analysis “adenoma activity, invasiveness, and relapse” there are any definitions provided. What authors mean writing adenoma activity? Haw exactly invasive growth was evaluated? How the authors define relapse (what criteria were used for variable clinically functioning PitNETs and nonfunctioning tumors??).
4. The article is difficult to follow due to large number of tables (13 tables with results). Vast majority of the results presented in tables are not significant. I’d strongly suggest present only significant results in one-two tables in main text and move all results that did not cross significance threshold to supplementary data. Similarly Figure 1 and 2 does not have to be presented - they present no difference.
5. A limitation of the study is that multiple statistical test were used. Many the results of many analysis only slightly cross significance threshold (as in analysis of control and invasive tumor groups for TEP1 rs1713418 and TERT rs2736098. I seem that any correction for multiple testing was applied.
6. Discussion section should be reedited Please, do not repeat description of the results in discussion section. Potential clinical implication should be discussed.
Minors
According to recent recommendations the term pituitary endocrine tumor (PitNET) is preferred over pituitary adenoma
I’d suggest proofreading the manuscript. The examples of writing imperfections are:
Lines 123 “normality difference” probably normal distribution
Line 133 134 These two sentences seem to have the same mining: “To test the statistical hypotheses, we chose a significance level of 0.05. A statistically significant difference was found when the p-value was 0.05.”
Line 326 “Table 9 “ seem to be mistaken reference
Typographic errors have to be corrected (example “Mano-Whitney U test line 132)
Author Response
Dear Editor and Reviewers,
We kindly appreciate the revision of our manuscript. We have highlighted the changes we made in the manuscript by using the track changes mode in MS Word. Hope that the revised manuscript will be acceptable for publication in your journal. Enclosed please also find attached our point-by-point response to the comments raised by the reviewers (editors).
Comments
- The main concern with this study is that it is focused on pituitary tumors as homogenous group. In fact particular subtypes of PitNEts are notably different in clinical manifestation, and molecular pathogenic mechanism including completely different mutations profile. Vast majority of the current and recently published studies on pituitary tumors take classification of PitNETs into account, which is definitely justified. 
Dear Reviewer, the main study concern is pituitary adenoma, not all pituitay tumors. Pituitary adenomas as a homogenous group is grouped regarding its hormonal activity, recurance, and invasivness.
- It is completely unknown what are diagnosis of the patients included in this study. Ay patients clinical /diagnostic characteristics are presented. Without this information it is unclear how to generalize the results of this study.
It was described in our previous publications. References were added.
- The following parameters were included in the analysis “adenoma activity, invasiveness, and relapse” there are any definitions provided. What authors mean writing adenoma activity? Haw exactly invasive growth was evaluated? How the authors define relapse (what criteria were used for variable clinically functioning PitNETs and nonfunctioning tumors??).
Hormonal acitivity, recurrence and invasiveness were described in our previous publications. References were added.
- The article is difficult to follow due to large number of tables (13 tables with results). Vast majority of the results presented in tables are not significant.  I’d strongly suggest present only significant results in one-two tables in main text and move all results that did not cross significance threshold to supplementary data. Similarly Figure 1 and 2 does not have to be presented - they present no difference.
Thank you for your suggestion. Tables and figures with non-significant results were moved to supplementary meterial.
- A limitation of the study is that multiple statistical test were used. Many the results of many analysis only slightly cross significance threshold (as in analysis of control and invasive tumor groups for TEP1 rs1713418  and TERT rs2736098. I seem that any correction for multiple testing was applied.
We only analysed binary logistic regresion, not multiple logistic regression. You are right, no correction for was applied. Bonferoni correction could be applied ig needed.  
Minors  
According to recent recommendations the term pituitary endocrine tumor (PitNET) is preferred over pituitary adenoma
Thank you for your suggestion, but we prefer to call it pituitary adenoma.
I’d suggest proofreading the manuscript. The examples of writing imperfections are:
Lines 123 “normality difference” probably normal distribution
It was corrected.
Line 133 134 These two sentences seem to have the same mining: “To test the statistical hypotheses, we chose a significance level of 0.05. A statistically significant difference was found when the p-value was 0.05.”
It was corrected.
Line 326 “Table 9 “ seem to be mistaken reference
It was corrected.
Typographic errors have to be corrected (example “Mano-Whitney U test line 132)
It was corrected.
Reviewer 2 Report
Paper is well written and interesting, look at these points to improve it:
- Results section is very long and sometimes not very incisive. Please revise and resume it.
- In results section there are several tables. Are all of them necessary?
- Lines 428-434. Pasqualetti et al. reported the role of blood cells in clinical oncology as potential biomarkers that are associated with survival. Discuss this point in the discussion section. Refs: doi: 10.3390/genes13061054 . Authors compared serum TEP1 levels between groups and genotypes for all polymorphisms.
- Line 448. "TERC rs35073794 may be associated with PA development". Conclusion is very short. Please improve it. What this paper add new to the literature?
Author Response
Dear Editor and Reviewers,
We kindly appreciate the revision of our manuscript. We have highlighted the changes we made in the manuscript by using the track changes mode in MS Word. Hope that the revised manuscript will be acceptable for publication in your journal. Enclosed please also find attached our point-by-point response to the comments raised by the reviewers (editors).
- Results section is very long and sometimes not very incisive. Please revise and resume it.
Results was devided by sub-titles. Also, non-significant results were moved to supplementary material.
- In results section there are several tables. Are all of them necessary?
Results section were updated. Non-significant results were moved to supplementary material.
- Lines 428-434. Pasqualetti et al. reported the role of blood cells in clinical oncology as potential biomarkers that are associated with survival. Discuss this point in the discussion section. Refs:  doi: 10.3390/genes13061054 . Authors compared serum TEP1 levels between groups and genotypes for all polymorphisms.
Dear reviewer, Pasqualetti et al. reported the role of blood cells in clinical oncology as potential biomarkers that are associated with survival. The study is not investigating Telomere associated genes, nor TEP1 levels. I am not sure if its really should be discussed in our paper if we have no survival data?
- Line 448. "TERC rs35073794 may be associated with PA development". Conclusion is very short. Please improve it. What this paper add new to the literature?
It was corrected.
Our study includes SNPs that have never been tested with pituitary adenoma before. It suggest TERC rs3507394 as a potential biomarker for PA.
Reviewer 3 Report
Gedvilaite et al determined the relationship between the molecular markers of telomerase complex (TERC rs12696304, rs35073794, TEP1 rs1713418, rs1760904, and TERT rs2736098, rs401681) and the relative leukocyte telomeres length with the development of pituitary adenoma. Overall, the study is well performed, but there are some minor points the author would need to address.
Minor points:
1. Results part need to be divided by sub-titles.
2. Could you detect TERC or TERT from serum? How are these associated with PA development?
Author Response
Dear Editor and Reviewers,
We kindly appreciate the revision of our manuscript. We have highlighted the changes we made in the manuscript by using the track changes mode in MS Word. Hope that the revised manuscript will be acceptable for publication in your journal. Enclosed please also find attached our point-by-point response to the comments raised by the reviewers (editors).
- Results part need to be divided by sub-titles.
Results was devided by sub-titles.
- Could you detect TERC or TERT from serum? How are these associated with PA development?
Yes, it could be detected from serum. However, we do not have enough samples anymore.
Reviewer 4 Report
In this manuscript, authors aim to correlate TEP1, TERC, and TERT genetic variants detected in blood samples with the development of PAs in a large cohort of patients.
General:
This manuscript is rather difficult to read and understand (e.g., the correct English expression for neoplasms of the anterior pituitary gland that do not produce hormones at clinically relevant levels is "non-functioning" PAs). As a consequence, professional English language assistance is recommended. Please explain all abbreviations at their first occurence in the text.
Methods section:
a) Highly important patient characteristics remain unclear: How did the patients present clinically? How was hormonal activity assessed? How was tumor invasiveness assessed? How many patients underwent medical treatment, surgery or irradiation? How were the clinical courses after initiation of treatment? How was tumor recurrence assessed?
b) Was it actually blood samples the authors obtained from the patients and from the healthy controls, or was it something else? The reviewer can only guess here.
c) This appears to be a comparison of TERC and TERT genotypes, and of TEP1 expression between patients with PAs and healthy controls as measured in a specimen taken from each individual at a single point in time. Since genotypes and protein expression levels in specimens may considerably vary over time, particularly in patients treated for a neoplastic disease, the following question arises: At what stage of the patients' disease (i.e., prior to medical treatment, during medical treatment, after medical treatment, prior to surgery, intraoperatively, after surgery, prior to irradiation, during irradiation, after irradiation, ...) were the samples taken?
c) Authors state: "A statistically significant difference was found when the p-value was 0.05." This appears rather unusual. Usually, p values less than 0.05 are considered statistically significant. Please clarify.
Conclusion:
Authors state: "TERC rs35073794 may be associated with PA development." The reviewer must recommend to moderate this statement, and to change the title of the manuscript accordingly, since, logically, measuring something at a single point in time does not allow one to state anything about "development".
Author Response
Dear Editor and Reviewers,
We kindly appreciate the revision of our manuscript. We have highlighted the changes we made in the manuscript by using the track changes mode in MS Word. Hope that the revised manuscript will be acceptable for publication in your journal. Enclosed please also find attached our point-by-point response to the comments raised by the reviewers (editors).
General:
This manuscript is rather difficult to read and understand (e.g., the correct English expression for neoplasms of the anterior pituitary gland that do not produce hormones at clinically relevant levels is "non-functioning" PAs). As a consequence, professional English language assistance is recommended. Please explain all abbreviations at their first occurence in the text.
Methods section:
- a) Highly important patient characteristics remain unclear: How did the patients present clinically? How was hormonal activity assessed? How was tumor invasiveness assessed? How many patients underwent medical treatment, surgery or irradiation? How were the clinical courses after initiation of treatment? How was tumor recurrence assessed?
Hormonal acitivity, recurrence and invasiveness were described in our previous publications. References were added. The samples were taken before medical treatment, surgery or irradiatio, thus we have no information about it.
- b) Was it actually blood samples the authors obtained from the patients and from the healthy controls, or was it something else? The reviewer can only guess here.
Blood samples were used for the DNA extraction. This information was added into the methods section.
- c) This appears to be a comparison of TERC and TERT genotypes, and of TEP1 expression between patients with PAs and healthy controls as measured in a specimen taken from each individual at a single point in time. Since genotypes and protein expression levels in specimens may considerably vary over time, particularly in patients treated for a neoplastic disease, the following question arises: At what stage of the patients' disease (e., prior to medical treatment, during medical treatment, after medical treatment, prior to surgery, intraoperatively, after surgery, prior to irradiation, during irradiation, after irradiation, ...) were the samples taken?
Samples were taken prior to medical treatment.
- c) Authors state: "A statistically significant difference was found when the p-value was05." This appears rather unusual. Usually, p values less than 0.05 are considered statistically significant. Please clarify.
Corrected: A statistically significant difference was found when the p-value was less than 0.05.
Conclusion:
Authors state: "TERC rs35073794 may be associated with PA development." The reviewer must recommend to moderate this statement, and to change the title of the manuscript accordingly, since, logically, measuring something at a single point in time does not allow one to state anything about "development".
The title was changed and the conclusion was corrected.
Round 2
Reviewer 4 Report
The text of the manuscript is still very hard to understand.
The reviewer suggests to explain in brief how tumor invasiveness, hormonal activity and recurrence were assessed, instead of only referencing previous studies conducted by the same group.
Obtaining information on patients' clinical courses is an additional effort for the authors. Such an effort will probably significantly enhance the quality of the manuscript.
Please have the statements as to the "development" of PAs moderated in the short summary, abstract, and manuscript, since the study design, assessing blood samples taken at a single point in time, does not allow one to draw any conclusions as to the "development" of these tumors.
The reviewer considers English language assistance with this manuscript mandatory.
Author Response
Thank you for your comment. We have made some revision in response to your comments
The text of the manuscript is still very hard to understand.
The manuscripts writing style was improved by native-English speaker.
The reviewer suggests to explain in brief how tumor invasiveness, hormonal activity and recurrence were assessed, instead of only referencing previous studies conducted by the same group.
The information was added to the manuscript.
Obtaining information on patients' clinical courses is an additional effort for the authors. Such an effort will probably significantly enhance the quality of the manuscript.
The information on PA subgroups was added.
Please have the statements as to the "development" of PAs moderated in the short summary, abstract, and manuscript, since the study design, assessing blood samples taken at a single point in time, does not allow one to draw any conclusions as to the "development" of these tumors.
This case-control study was conducted at the Lithuanian University of Health Sciences (LUHS), Kaunas, Lithuania 2016-2022. The research protocol was approved by the Kaunas Regional Biomedical Research Ethics Committee. Blood samples was not collected at a single point in time. Samples were colected for 6 years and stored at -20 celcius degree when DNA was extracted. This is why we think its allowed to state that TERC is likely to be associated with pituitary adenoma development.
The reviewer considers English language assistance with this manuscript mandatory.
The manuscripts writing style was improved by native-English speaker.
Round 3
Reviewer 4 Report
-